# Visualizations of Projected Rainfall Change in the United Kingdom: An Interview Study about User Perceptions

**Astrid Kause [1,2,3,\*], Wändi Bruine de Bruin [4] , Fai Fung [5], Andrea Taylor [1,2] and Jason Lowe [2,5]**

[1]  Centre for Decision Research, Leeds University Business School, Maurice Keyworth Building, University of Leeds, Leeds LS2 9JT, UK; A.L.Taylor@leeds.ac.uk

[2]  Priestley International Centre for Climate, School of Earth and Environment, University of Leeds, Leeds LS2 9JT, UK; jason.lowe@metoffice.gov.uk

[3]  Harding Center for Risk Literacy, Max Planck Institute for Human Development, Lentzeallee 94, 14195 Berlin, Germany

[4]  Sol Price School for Public Policy, Dornsife Department of Psychology, Schaeffer Center for Health Policy and Economics, and Center for Economic and Social Research, VPD 512D, 625 Downey Way, Los Angeles, CA 90089, USA; wandibdb@usc.edu

[5]  UK Met Office, Fitzroy Road, Exeter, Devon EX1 3PB, UK; fai.fung@metoffice.gov.uk

\*  Correspondence: a.kause@leeds.ac.uk; Tel.: +44-113-343-2865

**Abstract:** Stakeholders from public, private, and third sectors need to adapt to a changing climate. Communications about climate may be challenging, especially for audiences with limited climate expertise. Here, we study how such audience members perceive visualizations about projected future rainfall. In semi-structured interviews, we presented 24 participants from climate-conscious organizations across the UK with three prototypical visualizations about projected future rainfall, adopted from the probabilistic United Kingdom Climate Projections: (1) Maps displaying a central estimate and confidence intervals, (2) a line graph and boxplots displaying change over time and associated confidence intervals, and (3) a probability density function for distributions of rainfall change. We analyzed participants' responses using "Thematic Analysis". In our analysis, we identified features that facilitated understanding—such as colors, simple captions, and comparisons between different emission scenarios—and barriers that hindered understanding, such as unfamiliar acronyms and terminology, confusing usage of probabilistic estimates, and expressions of relative change in percentages. We integrate these findings with the interdisciplinary risk communication literature and suggest content-related and editorial strategies for effectively designing visualizations about uncertain climate projections for audiences with limited climate expertise. These strategies will help organizations such as National Met Services to effectively communicate about a changing climate.

**Keywords:** climate change; visualizations; uncertainty; risk communication; climate hazards

## 1. Introduction

Rainfall, storms, and heat waves are expected to become more frequent and intense as a result of climate change [1]. Stakeholders from public, private, and third-sector organizations as well as members of the general public face decisions about how to adapt to these changes [2,3]. To make informed decisions, they need reliable and understandable information about climate change and associated uncertainties [4].

Therefore, national governments in countries such as Australia or Switzerland aim to communicate probabilistic climate projections [5,6]. Similarly, in the United Kingdom, the Met Office UK produced

probabilistic climate projections about how the UK climate is expected to change over the course of the current century for the UK Climate Projections 2009 (UKCP09) and 2018 (UKCP18) platform [7,8]. These climate projections are simulations of how, for example, precipitation or temperatures might change in response to increasing concentrations of greenhouse gas emissions in the atmosphere, as well as how likely those changes are.

However, climate projections may be challenging for stakeholders from public, private, and third-sector organizations and members of the general public [2,4,9–11]. One potential difficulty in communicating climate projections is their inherent uncertainty [12]. This uncertainty arises from different sources, including natural variability in climate data resulting in imprecision, and uncertainty about how future human activities will impact on the climate, represented through different emission scenarios, or model uncertainty [13–18]. Thus, audiences' interpretations of climate projections and associated uncertainties may differ from what climate experts aim to communicate [19,20]. In addition, presenting uncertainty in climate information may delay public action on climate change [21], promote a "wait-and-see" approach [22], or increase public polarization about climate change [23]. This calls for communication formats that allow stakeholders to make adequate adaptation decisions and to integrate uncertainties into their decisions [24].

### 1.1. Visualizations for Communicating Uncertain Climate Projections

Research from the cognitive and behavioral sciences suggests that, compared to text or number formats, visualizations can help individuals to better understand information about complex issues, such as climate change [11,25]. However, most evidence about perceptions of visualizations comes from the health domain, where these have been used to improve individuals' understanding about risks [11,26–30] and to promote self-protective behaviors [26]. However, findings from the domain of health may not always transfer to climate because climate projections may be more complex, more inherently uncertain, occur across longer time periods, and have consequences that vary widely between individuals or institutions.

In the climate domain, uncertainties make the design of visualizations challenging [11,24]. For example, uncertainties in climate projections may result in a large number of data points, increase possible relations between displayed variables, and may leave it open as to how a hazard will develop in the future [11].

Here, we focus on three types of visualizations that are commonly used to communicate climate projections and associated uncertainties. These were designed for the United Kingdom Climate Projections (UKCP) platform [7,8]. First, maps [12] can show geographical variations in both relative change in a climate hazard as well as uncertainty. A set of separate maps can be used for communicating regional projections, with each reflecting different probabilities of occurrence. For example, based on recent evidence about how to design maps of climate projections [12], the UK Met Office redesigned maps from UKCP09 for UKCP18 (Figure 1). Uncertainty resulting from model disagreement has also been represented through patterns, such as hatching, stippling, or color [12]. Alternatively, change and associated uncertainty have also been displayed in two separate maps [30].

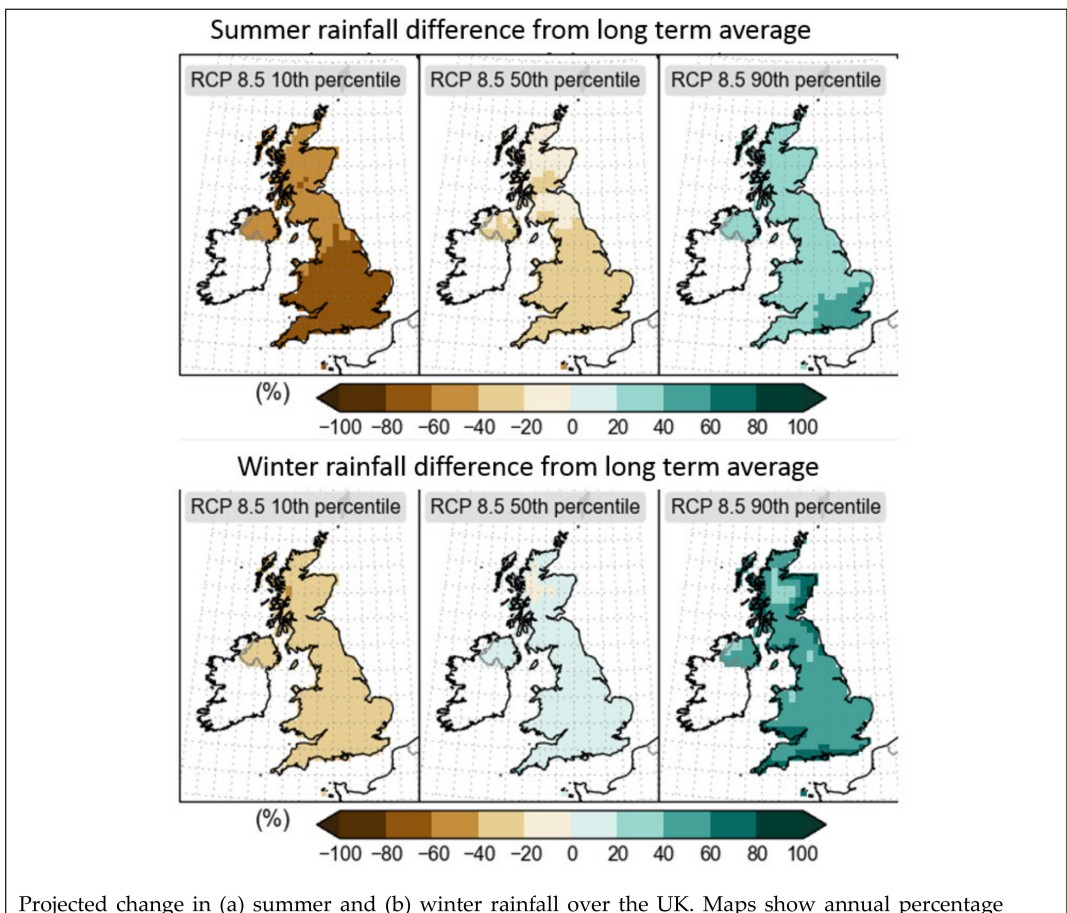

Projected change in (a) summer and (b) winter rainfall over the UK. Maps show annual percentage differences from the long-term (1981–2010) average and the average between 2061 and 2080. The projections are probabilistic: The 50th percentile is the central estimate; values are very unlikely to be greater than the 90th percentile or less than the 10th percentile. The maps are for a "business-as-usual" scenario where emissions continue unabated.

**Figure 1.** Map and associated caption displaying summer and winter rainfall change in the United Kingdom, as prepared for Met Office communications and presented to participants. Note: In the interviews, participants saw these maps with their accompanying captions, both of which were prototypes prepared by the UK Met Office for the United Kingdom Climate Projections 2018 (UKCP18) [7]. Participants did not receive any additional text or explanation.

Second, line graphs: These also tend to be used to show changes over time, with each line representing a single estimate over a period of time [9]. They may also include separate lines to reflect a confidence interval around the line for the central estimate. For example, the Met Office designed such a line graph for UKCP09 (see Figure 2).

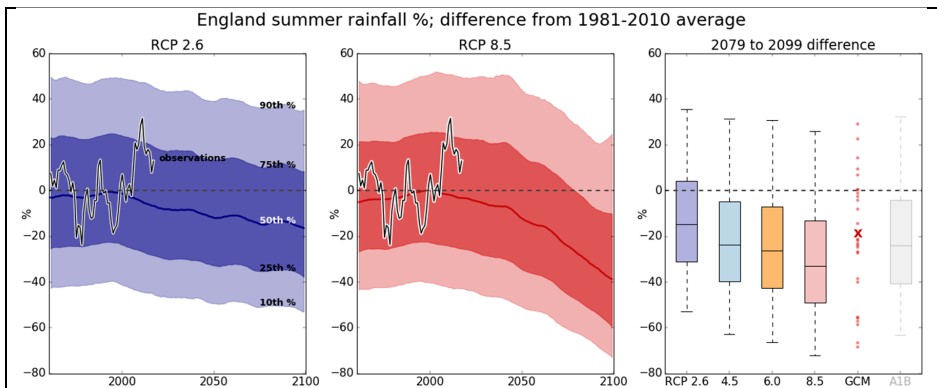

Projected change in summer rainfall for England. Graphs (a) and (b) show annual percentage differences from the long-term (1981–2010) average for years between 1961 and 2100. The projections are probabilistic: The 50th percentile is the central estimate; values are very unlikely to be greater than the 90th percentile or less than the 10th percentile. A black line shows precipitation using data from climate projection models. **(a)** shows projections using a scenario that seeks to aggressively limit emissions of greenhouse gases (RCP 2.6). **(b)** shows projections for a "business-as-usual" scenario where emissions continue unabated. **(c)** shows the projected range of percentage change in rainfall between the long-term average and the 2070–2099 average. The boxplots show the probabilistic projections under four scenarios: RCP 2.6, 4.5, 6.0, and 8.5, as well as the SRES A1B used in the United Kingdom Climate Projections 2009. The red dots show a range of outputs from a combination of global climate models; the red cross shows the mean of the estimates.

**Figure 2.** Line graphs and boxplots, and associated caption displaying summer rainfall change in England, as prepared for Met Office communications and presented to participants. Note: In the interviews, participants saw these line graphs, boxplots and captions, which were prototypes prepared by the UK Met Office for the United Kingdom Climate Projections 2018 (UKCP18) [7]. Participants did not receive any additional text or explanation. RCP: Representative Concentration Pathway representing scenarios of different concentrations of greenhouse gas emissions in the atmosphere. GCM: Global Climate Models. SRES A1B: Special Report on Emissions Scenarios set A1B from the Intergovernmental Panel on Climate Change.

Third, probability density functions (PDFs) may be used to present the likelihood of specific levels of change in precipitation or temperature. They show different amounts of change in a climate hazard, as well as their respective likelihoods for a single point in time and a specific geographical area. For example, the Met Office designed such a PDF for UKCP09 and UKCP18 (Figure 3).

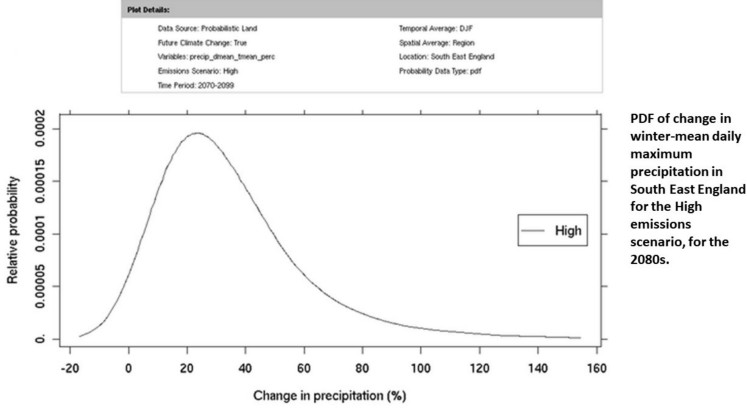

**Figure 3.** Probability density function displaying precipitation change in South-East England, as used in Met Office communications and presented to participants. Note: In the interviews, participants saw this probability density function and the accompanying caption, which were designed by the UK Met Office for the United Kingdom Climate Projections 2009 (UKCP09) [8] and later adapted for UKCP18 [7]. Participants did not receive any additional text or explanation.

Communications about climate projections should be tested with members of the intended target audience, such as stakeholders in local governments or water management, who make decisions for adapting to a changing climate [31]. Such communication materials are often produced with the general public in mind [11]. They aim to follow best practices for designing understandable communications, similar to those developed for health communications [25,32]. This is important because one potential concern is that communications about climate projections may be difficult for intended target audiences to understand [31].

### 1.2. The Current Study

In qualitative, semi-structured interviews with stakeholders from public, private, and third-sector organizations in the UK, we presented a set of maps (Figure 1), a set of line graphs and boxplots (Figure 2), and a probability density function (Figure 3) with accompanying captions that illustrated different aspects of projected future rainfall in the UK. Participants were asked to discuss any visualization features that they felt helped or hindered their understanding. We aimed to answer the following research questions: (1) What are the most commonly mentioned features facilitating understanding? and (2) What are the most commonly mentioned barriers to understanding? We also discuss strategies identified both from findings and the wider risk communication literature about how to overcome those barriers.

## 2. Materials and Methods

### 2.1. Participants

We recruited 24 participants through the United Kingdom Climate Projection user networks of the Met Office UK, as well as networks of the Universities of Leeds and Oxford. Participants were professionals faced with climate adaptation decisions in a variety of settings: County council committees for infrastructure planning, organizations advising the UK government and major engineering companies on climate change, charities taking care of nature reserve and wildlife preservation, and water companies. We recruited participants from the different target audiences that UKCP communications aim to address (see Table 1). Such diverse samples are also recommended to increase the likelihood of identifying potential misunderstandings in the interviews [33]. Our sample size was larger than the 10–15 needed to reach "saturation" on identifying the most common misunderstandings [33]. Table 1 describes participant characteristics.

**Table 1.** Participant characteristics.

| N | 24 |
|---|---|
| Sector | Water infrastructure: 7 (29%) <br> Public infrastructure: 8 (33%) <br> Nature reserve protection (charity): 2 (8%) <br> Consultancy: 4 (17%) <br> Communications: 1 (4%) <br> Climate science: 1 (4%) <br> Building/construction: 1 (4%) |
| Years of expertise within area of work | M = 14.02 (SE = 2.15; Range: 1–41) |
| Familiarity with United Kingdom Climate Projections (1–7 scale) | M = 4.38 (SE = 0.36; Range: 1–7) |
| Importance of United Kingdom Climate Projections for planning (1–7 scale) | M = 4.79 (SE = 0.45; Range: 1–7) |
| Previous use of visual climate communications | No: 29%; Yes: 71% |
| Gender | 63% male |

Note: Familiarity was assessed by asking "On a scale from 1 to 7, 1 meaning not familiar at all and 7 meaning very familiar, how familiar are you with Met Office climate projections?" Importance was assessed by asking "On a scale from 1 to 7, 1 meaning not important at all and 7 meaning very important for your work, how important are Met Office climate projections?".

## 2.2. Interview Protocol and Materials

The interview consisted of two parts. In the first part, participants were asked how they use climate information in their work (see Table 1). We also assessed their familiarity with and perceived importance of the United Kingdom Climate Projections issued by the UK Met Office (see Table 1). In the second part, participants viewed three visualizations and accompanying captions, designed by the UK Met Office. Visualizations and captions were (1) a set of map prototypes prepared for UKCP18 (Figure 1) [7], (2) a set of line graph and boxplot prototypes prepared for UKCP18 (Figure 2) [7], and (3) a probability density function taken from the UKCP09 user interface (Figure 1) [8]. The choice of visualizations is suitable, because maps, time series, and probability density functions are commonly used in communications of climate projections, including UKCP09 and UKCP18 [7,8]. The visualizations prepared for UKCP18 were included so that our findings could inform their use in the UKCP18 report. Participants did not receive any other information. Although the user interface also presents some general guidance about how to use climate projections [34,35] and about Representative Concentration Pathways (RCPs) (https://www.metoffice.gov.uk/binaries/content/assets/metofficegovuk/pdf/research/ukcp/ukcp18-guidance---representative-concentration-pathways.pdf), this guidance was not provided here. Visualizations and associated captions were presented on their own without the main text, because recommendations in the communication literature state that figures and captions should be understandable without having to read the main text [36]. We tested prototypes, so that any issues identified in the interviews could potentially be addressed in the version to be used in Met Office communications They thus still included terminology such as "business-as-usual" for describing high-emission scenarios, which was criticized recently by other authors [37]. We chose to present visualizations about projected precipitation, because precipitation in particular is expected to become more extreme in the UK, and is central to planning by UK practitioners and policy makers [7]. We refrained from editing visualizations before presenting them in the interviews. This was done to maximize empirically validated insights for regarding the design of such visualizations, including content-related issues as well as issues related to editing.

When viewing each visualization, participants were first asked to "think aloud" while describing their interpretations of it [38]. In a "marked protocol" [38], participants were prompted with a red rectangle to focus and comment on the visualization first, and then on any accompanying caption. They were then asked to interpret the visualizations. For example: "What change would you predict for the Isle of Man?" and "What change would you predict for the area of greater London?" when viewing the map, "Can you please describe how much precipitation will change on average in England in the year 2025?" when viewing the line graph and boxplots, and "How much will precipitation change according to this graph?" when viewing the probability density function. After interpreting each visualization and associated caption, participants rated how understandable and how helpful they found each of these, on scales from 1 (not at all) to 7 (very). They were also asked to explain each of these two ratings and how they would improve visualizations and captions, if at all. The two ratings for each of the visualizations and each of the captions were strongly correlated (all r's > 0.57, all p's < 0.01). We thus averaged those separately for each visualization and caption into "overall helpfulness" ratings. Due to the small sample size, these "overall helpfulness" ratings need to be interpreted with care.

The full interview protocol is provided in Table S1 of the Supplementary Information. Interviews lasted 50–90 min. Upon completion, each participant received an online shopping voucher worth £75. The ethical review board of the University of Leeds approved the interview study (LTLUBS-216).

## 2.3. Data Analysis

All interviews were audio-recorded and transcribed verbatim. The first author (AK) coded all interviews using the software NVivo 11 and 12 [39]. For each interview section, she used a coding framework based on "Thematic Analysis". Thematic Analysis aims to identify common patterns or meanings in participants' answers in response to specific research questions [40]. This identified features

that facilitated understanding, as well as barriers occurring when participants tried to understand each of the three visualizations [38]. To answer research questions 1 and 2, emerging overarching themes were identified and coded separately for each type of visualization (Tables 2–4). We assessed in one-sample *t*-tests whether quantitative "understandability" and "helpfulness" ratings were relatively different from scale midpoints. As the visualizations presented here have fundamentally different aims, vary in the projections they communicate, and differ in many other aspects, such as displayed seasons and emissions scenarios, we refrained from conducting statistical comparisons between them. For captions, we additionally calculated Flesch–Kincaid reading-level grade statistics, a general indicator of the reading comprehension level needed to understand the presented text [41].

## 3. Results

### 3.1. Sample Response to MAPS

On average, participants rated the maps as relatively helpful (Figure 4), seen in mean "overall helpfulness" ratings being higher than the scale midpoint (M = 4.49; SD = 1.30; t (23) = 3.72, *p* = 0.001). On average, participants also found the caption relatively helpful, with these overall ratings also being higher than the midpoint of the scale (M = 4.57, SD = 1.26, t (23) = 4.18, *p* < 0.001). Below, we discuss features of maps and captions that participants identified as helping their understanding (Table 2a) and as barriers to understanding (Table 2b).

**Table 2.** (**a**) Features of the map identified as facilitating understanding; (**b**) Features of the map identified as hindering understanding.

| (a) | | |
|---|---|---|
| **Feature** | **% of Participants** | **Example Quote** |
| *Map colors* | 71% | "Color-wise, yeah, it's interesting because I think, you know, when you're talking about with the brown and the green, it's very much kind of brown—you associate the color brown with drought and then you've got green which is lush, and that's how I've read that whether or not that's what the data is showing." (Participant 16) |
| *Readability* | 42% | "I think the first part of the text, the first three lines is fine. Very readable and understandable. It shows the difference. It gives you the years which the average has been generated from." (Participant 6) |
| *Presented emission scenario* | 25% | "The business-as-usual scenario where emissions continue unabated. That's very straightforward, I understand that. So, I understand that if we just carry on like we are, then these are the predicted scenarios" (Participant 17) |
| *Description of probabilistic estimates* | 21% | "The statement, 'Values are very unlikely to be created in the 90th percentile and less than the 10th percentile' is probably the most helpful bit." (Participant 4) |
| (b) | | |
| **Feature** | **% of Participants** | **Example Quote** |
| *Acronyms and jargon* | 83% | "I don't understand what the RCP 8.5 means. I can only assume that it means it's one of the projections that they're using." (Participant 3) |
| *Probabilistic estimates* | 79% | "It takes a lot of time to work out what each of those things mean and you actually have to understand what a percentile is in the first place, which I think a lot of people don't, including myself." (Participant 22) |
| *Size and resolution* | 71% | "It would be helpful if the figures were a little bit larger in order to see some of the detail." (Participant 18) |
| *Time periods* | 58% | "So, why the range, 2061 to 2080, and why was the average between that taken?" (Participant 15) |
| *Match between visualization and caption* | 50% | "There's actually no link between where it says RCP 8.5 on the graphs and then the 'business-as-usual'. That needs linking or being made a bit more clear. And if these are for general purpose use, RCP 8.5 won't mean anything, but 'business-as-usual' might." (Participant 2) |
| *Relative change* | 46% | "The difference from the long-term average. But then what does that actually mean in terms of how much dryer or how much wetter it is. It feels like it's—for a specialist audience this will mean something because they already know the background and they already know how much rainfall there is, and they would know how much 20% would really matter to various things." (Participant 5) |

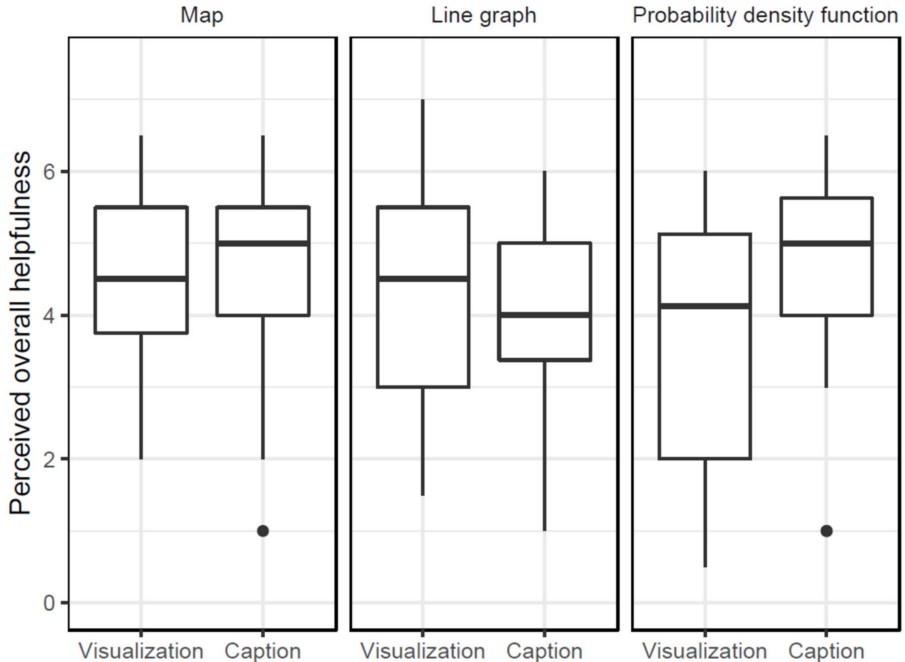

**Figure 4.** Boxplots displaying "overall helpfulness" ratings for each visualization and each caption. Note: Boxplots show the "overall helpfulness" ratings for each visualization and caption. Boxes represent the interquartile range (IQR). Whiskers represent the range between minimum and maximum values (IQR ± 1.5 * IQR). Dots represent outliers outside of this range. The lack of outliers towards the upper end of the scale suggests that participants had a tendency to give responses slightly above the scale mid-point (see statistical tests reported in Sections 3.1–3.3). Due to the small sample size, these ratings need to be interpreted with care.

### 3.1.1. Map Features That Facilitated Understanding

*Map colors.* Seventeen participants (71%) found the map colors easy to understand, and correctly interpreted blue shades as change towards more rainfall and brown shades towards less rainfall. They also pointed out that blue and brown could be discriminated by color-blind users.

*Readability.* Ten (42%) participants liked the readability of the first two sentences in the caption: "Projected change in (a) summer and (b) winter rainfall over the UK. Maps show annual percentage differences from the long term (1981–2010) average and the average between 2061 and 2080." For these two sentences, the Flesch–Kinkaid reading-level grade statistic was 8.73, which suggests that they required reading comprehension skills at the 8th–9th grade level [41].

*Presented emission scenario.* Six participants (25%) further underlined that the business-as-usual emission scenario ("Representative Concentration Pathway 8.5" (RCP 8.5)) described in the graph header was easy to understand, perhaps in part because this was the only scenario displayed. This scenario represented a rising trajectory of greenhouse gas emission concentrations through the 21st century [7].

*Description of probabilistic estimates.* Five participants (21%) mentioned that they found the description of probabilistic estimates in the caption helpful. Specifically, they referred to the sentence: "The projections are probabilistic: The 50th percentile is the central estimate, values are very unlikely to be greater than the 90th percentile and less than the 10th percentile." One participant interpreted the central estimate as the most likely value, others mentioned that the description would help them communicate to others what a probabilistic estimate is.

### 3.1.2. Map Features That Hindered Understanding

*Acronyms and jargon.* Twenty participants (83%) were confused about the use of unfamiliar acronyms and terminology (Table 2b). One unfamiliar acronym was "RCP", which stands for "Representative Concentration Pathway"—an emission scenario based on a specific socio-economic future development. They were also unfamiliar with the phrase "unabated emissions" as well as statistical terms such as "percentile" and "probabilistic".

*Probabilistic estimates.* Nineteen participants (79%) mentioned that probabilistic estimates and their explanations in the caption may be hard to interpret, either by themselves or by individuals with whom they interact professionally. Across the three maps, probabilistic estimates were communicated as central precipitation estimates with lower and upper bounds, representing 10% and 90% thresholds. Several participants noted the large range represented through these probabilistic estimates, which included both drier and wetter winters. They also noted that ranges were larger for summer than for winter. They therefore asked for a more elaborate explanation of probabilistic estimates.

*Size and resolution.* Seventeen participants (71%) mentioned that the size and resolution of the map and the font size of the caption were too small. Specifically, they had difficulties seeing specific geographic areas, such as the small islands in the Northwest of Scotland. The coastline was also hard to see because of the relatively thick black coastal borderline that was used in the map.

*Time periods.* Fourteen participants (58%) described misunderstandings about the time periods represented in the map. Specifically, the caption described the map as showing "annual percentage difference from the long-term (1981–2010) average and the average between 2061 and 2080." Participants wanted to know why these specific time periods were chosen. They also expressed confusion about the different lengths of the future and past time periods, as well as the different wordings used to describe them. Several participants even interpreted the two time periods as two projections instead of as a comparison of the future against a historical baseline.

*Match between visualization and caption.* A total of 12 participants (50%) seemed confused about how terms used in the caption related to information in the visualization. For example, the emission scenario was described in the map header as "RCP 8.5", and in the accompanying caption as "'business-as-usual scenario' where emissions continue unabated." They also found the letters "(a)" and "(b)" in the caption confusing and asked how those linked to the visualization, which did not show equivalent letters. Reactions to the legend below the maps were mixed. In intervals of 10 percentage points, these specified amounts of change. Amounts were represented through different shades and colors. Eight participants (33%) struggled to match the color shades in the scale with the map or to recognize the associated numerical percentage intervals. As a result, when asked to read relative change in a specific location of the maps, they frequently provided ranges of change (such as "10–30%") instead of point estimates. They also suggested using brighter colors to make them easier to differentiate on the scale, or to add labels to the map indicating the amount of change displayed in a region of interest.

*Relative change.* Almost half of participants (46%) noted that baselines were missing for the presented amount of change in rainfall. They explained that the presented amount of change in rainfall was difficult to interpret without the presentation of a baseline. They also noted that baselines likely varied, affecting how they would interpret relative change across regions. Confusion was expressed about changes that were negative and changes that potentially exceeded 100%. Furthermore, participants wondered whether the maps presented changes compared to an overall yearly average or to an average for each season, as well as what precipitation change would look like in terms of volume, intensity, or duration of precipitation.

### 3.2. Sample Response to Line Graphs and Boxplots

On average, participants found the line graphs and box plots relatively helpful, seen in "overall helpfulness" ratings being higher than the midpoint of the scale (M = 4.28, SD = 1.55, t (23) = 2.48, $p$ = 0.02)). On average, participants also found the caption relatively helpful, with these overall ratings also being higher than the midpoint of the scale (M = 4.11, SD = 1.27, t (23) = 2.38, $p$ = 0.03; Figure 4).

Below, we discuss the features of this visualization and its associated caption, which participants identified as facilitating their understanding (Table 3a) and as barriers to their understanding (Table 3b).

**Table 3.** (**a**) Features of the line graphs and boxplots identified as facilitating understanding. (**b**) Features of the line graphs and boxplots identified as hindering understanding.

| (a) | | |
|---|---|---|
| **Feature** | **% of Participants** | **Example Quote** |
| *Readability* | 42% | "The first sentence is clear, projected change of summer rainfall for England, perfectly clear, tells you what it is. The text does tell you all the information that you are looking at." (Participant 9) |
| *Comparison of different emission scenarios* | 29% | "The bit that says (a) [the left panel in Figure 2] shows projections using the scenario that limits emissions. (b) [the middle panel in Figure 2] shows projections for business as usual." (Participant 8) |
| *Colors and shading* | 25% | "There's a 50th percentile line, which vaguely fits the squiggly line at the left-hand end, and then continues towards the right, and then there's some blocks of color, which are 25th, 75th, 10th, 90th percentile, dark blue, light blue. So, the dark blue bit is more likely to happen than the light blue bit, so that kind of makes sense." (Participant 4) |
| *Comparison of different emission scenarios (boxplots)* | 17% | "Where it [the right panel in Figure 2] is useful is showing the kind of broader range of changes or the difference rather in between long-term average and the baseline for a number of difference scenarios." (Participant 22) |
| (b) | | |
| **Feature** | **% of Participants** | **Example Quote** |
| *Acronyms and jargon* | 83 | "There's far too many abbreviations, far too many acronyms with no information on what those acronyms are. [ … ] I don't know what SRES A1B is." (Participant 4) |
| *Boxplots: Comparison of different emission scenarios* | 63 | "I don't know whether the box plots belong on these visualizations [ … ] because you're actually portraying something in a different way, and I think that's complicated." (Participant 2) |
| *Real observation data* | 50 | "On the text it says that the black line shows precipitation using data from climate protection models, but the label on the graph says observed data so I think that there needs to be a clarification on what that freeform black line is actually showing." (Participant 16) |
| *Probabilistic estimates* | 46 | "I miss out a bit when it says about probabilistic, because I struggle to understand it." (Participant 3) |
| *Amount of information* | 46 | "It's an overwhelming amount of information with not much clarity about what the three different boxes are showing." (Participant 15) |
| *Time periods* | 33 | "In 'summer rainfall percent difference from 1981–2010 average'. And this is over a time period presumably looking up to 2010. It's not clear where the starting point is, though one might infer it's 1981—that doesn't seem to fit with the scale." (Participant 10) |
| *Graph colors* | 29 | "The colors aren't explained properly at all to me. Whatever it is, there looks to be more red towards the end of the century and there's a bit more blue under a lower scenario, but I'm really not sure what that means at all." (Participant 2) |
| *Match between visualization and caption* | 25 | "In the text, it says Graphs A and B show annual percentage, but when you look at the graphs, you don't have A and B on them." (Participant 14) |
| *Relative change* | 25 | "The baseline itself is hard to understand, and then what A and B mean is set against not a meteorological baseline but against a political baseline, in effect." (Participant 2) |
| *Labels for probabilistic thresholds on the line graph for RCP8.5* | 21 | "It's obvious with the color change in the dark and the paler colors, but there isn't actually a key." (Participant 1) |

### 3.2.1. Line Graph and Boxplot Features Facilitating Understanding

*Readability.* Almost half of participants (42%) described the first two sentences of the caption ("Projected change in summer rainfall for England. Visualizations (a) and (b) show annual percentage differences from the long-term (1981–2010) average for years between 1961 and 2100.") as simple and easy to understand. For these two sentences, the Flesch–Kinkaid reading-level grade statistic was 9.64, suggesting that these sentences required reading comprehension skills at the 9th–10th grade level [41].

*Comparison of different emission scenarios.* A third of participants (29%) pointed out that they benefited from being presented with the different emission scenarios in the three graphs. Comparing these graphs facilitated their understanding about how different types of emission scenarios affected

local consequences, such as precipitation change. Some identified the boxplots as the easiest to understand, because it displayed several different emission scenarios. They correctly described that all displayed scenarios were associated with a decrease in precipitation.

*Colors and shading.* Six participants (25%) mentioned that the colors in the visualizations were straightforward to interpret. Precipitation change associated with a low-emission scenario was displayed in blue the line graph for RCP2.6. Precipitation change associated with a high-emission scenario was displayed in red in the line graph for RCP8.5. Some participants interpreted red as more and blue as less alarming. Others said that they found the shading helpful for understanding that lighter colors represented the 10th and 90th percentile, and darker colors the 25th and 75th percentile.

*Comparison of different emission scenarios (boxplots)).* Only 17% of participants found the boxplots in Graph (c) (Figure 2) helpful. They said that they benefited from comparing different emission scenarios and thus seeing the overall possible range of change.

### 3.2.2. Line Graph and Boxplot Features Identified as Hindering Understanding

*Acronyms and jargon.* Most participants (83%) were unsure about the meanings of acronyms used in the line graphs and boxplots, like RCP 2.6 and 8.5 in the line graphs, or the emission scenario SRES A1B and Global Climate Model (GCM) in the boxplots. They also expressed confusion about the terms "global climate model" and statistical terms like "probabilistic" and "likely" or "very unlikely" used in the caption.

*Boxplots: Comparison of different emission scenarios.* Fifteen participants (63%) referred to this graph as potentially confusing. It compares projected ranges of change in precipitation for different emission scenarios, as well as a set of projections from the Global Climate Model (GCM) and a set of emission scenarios A1B from the Intergovernmental Panel on Climate Change [7]. Specifically, participants mentioned that across graphs, the different projection time periods (2061–2100 in the line graphs versus 2079–2099 in) the boxplots and the different display modes made comparisons difficult. They were also unsure about how to interpret RCPs, or about how to compare them to other presented model outcomes. The meaning of the Global Climate Model also tended to be unfamiliar. Participants suggested to present the boxplots as a separate visualization and to either remove or better explain the Global Climate Model and the projections based on the A1B emission scenarios.

*Observation data.* Half of the participants were confused about the "black wiggly line" shown in the line graphs. This line represents observation data in the time period 1950–2018 and is described in the caption as "precipitation using data from climate projection models". While some participants conflated this line with the central model estimates in dark blue and dark red, or found it hard to relate the explanation in the caption ("data from climate projection models") to the graphs, others struggled to understand why the observation line exceeded the 75% range. Furthermore, it was unclear to several participants whether the line representing the central estimate was a UK-wide average.

*Probabilistic estimates.* Eleven participants (46%) mentioned that they found it difficult to understand probabilistic estimates expressed as percentiles. The line graphs displayed these as color bands with different values. The line graph for RCP 2.6 indicated numerical values for percentiles. Participants struggled to interpret the color bands representing probabilistic estimates in percentiles, as well as the terms "very unlikely" and "likely" used to describe probabilistic estimates in the caption. They also interpreted probabilistic estimates as communicating large variability in climate. Several participants requested a legend that would explain the differences in color value. Some also requested a legend explaining the boxplots.

*Amount of information.* Eleven participants (46%) stated that the three graphs were too much to digest. The line graphs display precipitation change estimates based on the highest and lowest greenhouse gas emission pathways (RCP 2.6 and RCP 8.5). A third graph, using boxplots, compares these two scenarios plus two more projection sets of estimates from different climate models to each other. Consequently, several participants suggested to present the line graphs together, and the boxplots separately. Only some identified the line graphs as precipitation change associated with the

lowest and highest greenhouse gas emission pathways (RCP 2.6 and RCP 8.5) and the boxplots as an overall comparison of all four emission scenarios. Others also said that they had to read the caption very carefully in order to understand the graphs, or jump back and forth between the graphs and the caption.

*Time periods.* Eight participants (33%) mentioned that they were confused about the time periods represented in the three graphs. For the line graphs, they found it difficult to compare the baseline (described as 1982–2010 in the caption) to the change displayed in the graph and the time periods on the x-axis. Here, they tried to find the starting point (1981) on the axis, which, due to the 50 year intervals, was not clearly marked. Regarding the boxplots, some noted that the change time period was different, and included only 20 years (rather than 150, as on the line graphs).

*Graph colors.* Seven participants (29%) struggled to interpret graph colors. This included that they did not know why the line graphs were blue and red. One participant would have preferred to see matching colors between scenarios displayed in the line graphs and those in the boxplots. Participants thus asked for a legend specifying what the color-coding on all three graphs meant.

*Match between visualization and caption.* Seven participants (29%) pointed out that different elements of the caption did not match on the three graphs shown. For example, the caption referred to these as (a), (b), and (c), but these labels were missing on the graphs. Furthermore, the caption described "model projections", but the graphs also included actual observations. As the caption described the latter as a "black line" rather than using the same term as in the graph ("observations"), only a few participants related the description in the caption to the line shown in the graphs.

*Relative change.* Six participants (25%) asked about the relative percentage change communicated on the y-axis of all three graphs. They suggested to better explain the y-axis itself, and to use communications of absolute change in the original unit. They struggled with translating time periods (such as 1981–2010) into baselines. Furthermore, some did not know how to relate the starting point of the baseline (1981) to the x-axis, which included a much longer time period.

*Labels for probabilistic threshold labels on the line graph for RCP8.5.* Five participants (21%) noticed that the labels relating the colors to probabilistic thresholds on Graph (a) (for the 25th, 50th, 75th, and 90th thresholds) were missing on Graph (b). They understood that thresholds in the line graph about RCP8.5 were likely equivalent to the thresholds in the line graph about RCP2.6, but noted that similar numerical labels would have made (b) easier to understand.

### 3.3. Sample Response to the Probability Density Function

The "overall helpfulness" rating was M = 3.86 (SD = 1.77), which was similar to the scale midpoint ($t$ (23) = 1.01, $p$ = 0.32)). The "overall helpfulness" rating of the caption and the plot details was M = 4.55 (SD = 1.51), which was higher than the scale midpoint ($t$ (22) = 3.35, $p$ = 0.003; Figure 4). Below, we discuss features that participants identified as helping their understanding (Table 4a) and as barriers to understanding (Table 4b).

**Table 4.** (**a**) Features of the probability density function identified as facilitating understanding. (**b**) Features of the probability density function identified as hindering understanding.

| (a) | | |
|---|---|---|
| **Feature** | **% of Participants** | **Example Quote** |
| *Caption and plot details* | 58% | "Most helpful, yeah, the kind of second half of it. So it gives you the region, the emissions scenario, and the time slice." (Participant 22) |
| *X-axis* | 38% | "This is clearly a PDF of different magnitudes of change. The x-axis is clear to understand what this is." (Participant 11) |
| (b) | | |
| **Feature** | **% of Participants** | **Example Quote** |
| *Relative probability of change* | 75% | "The relative probability, I do not understand." (Participant 6) |
| *Acronyms and jargon* | 75% | "Most people would think that means it's a PDF in the way that a document can be a PDF, whereas actually, it's probably meaning it's a probability density function." (Participant 2) |
| *Plot details* | 71% | "I think the plot details box looks a bit kind of clunky and I don't think it necessarily adds something to it. I'd rather see a little blurb underneath with emission, you know, that we're using the high-emission scenario, and I think in terms of the title, you know, it would be more appropriate to have a South East of England title up there, you know, looking at the high-emissions scenario and a little bit of blurb, so yeah, I'd remove that completely." (Participant 3) |
| *Visualization legend* | 38% | "I don't get why the key [legend] just says 'high' [ … ]. Is that the high-emission line? No, it's not because it's talking about precipitation. I don't know what the line of high is." (Participant 14) |

### 3.3.1. Features Facilitating Understanding of a Probability Density Function

For the probability density function, participants mentioned only two features that facilitated their understanding. Thus, they mentioned much fewer helpful features compared to the other visualizations.

*Caption and plot details.* Over half of the participants (58%) pointed out that the caption and plot details were easy to understand because they specified the emission scenario, the future time period, and South East England as the specific region the projection was for. While some considered the term "mean daily maximum precipitation" useful, others evaluated the term as precise but long. For the caption, the Flesch–Kinkaid reading-level grade statistic was 7.37, suggesting that these sentences required reading comprehension skills at the 7th-grade level [41].

*X-axis.* The x-axis, which described the amount of change in precipitation, was evaluated by nine (38%) participants as straightforward to understand. Some pointed out that this axis was easier to understand than the y-axis, which indicated the relative probability of change.

### 3.3.2. Features of the Probability Density Function Identified as Barriers to Understanding.

*Relative probability of change.* The most frequently mentioned barrier was the y-axis of the probability density function. The y-axis displays relative probabilities associated with different amounts of change. Eighteen (75%) participants struggled with interpreting small probability values and translating them in a meaningful way: This included the slopes, as well as the peaks shown in the distribution, and how those related to the y-axis. Several participants asked for a display in percentages, as well as for gridlines to assist in relating the axis to the curve shown.

*Acronyms and jargon.* Eighteen (75%) participants questioned the meaning of the (unexplained) acronyms. The most confusing acronym was "PDF" in the caption, which may be interpreted as the more commonly used "Portable Document Format". A few participants also pointed out that the acronym "DJF" (December, January, February) in the accompanying plot details was unfamiliar to them.

*Plot details.* Seventeen (71%) participants described the plot details on the top as too difficult to read due to their small size, and also described them as difficult to understand due to jargon and acronyms. Plot details list the type of variable shown, the emission scenario, time period, season, the

location and type of spatial average, and the plot type. Participants questioned the importance of different pieces of information for the overall message of the visualization.

*Visualization legend.* Nine participants (38%) questioned the meaning of the visualization legend, indicating that the projection was associated with a high-emissions scenario. As it was unclear that the "High" referred to an emissions scenario, participants asked for further explanation, or suggested the complete removal of the legend in its current format.

## 4. Discussion

Stakeholders in public, private, and third-sector organizations, as well as members of the general public, need to adapt to a changing climate. In order to inform their decisions, visualizations about climate hazards need to be designed to be understandable to those audiences. However, climate projections about hazards such as precipitation involve uncertainty and may be difficult to communicate. The current study represents a first step towards better understanding how such target audiences perceive commonly used climate projection visualizations.

In a series of semi-structured interviews, we studied how stakeholders from across the United Kingdom perceived three commonly used visualizations about uncertain climate projections that, in this case, were designed by the UK Met Office: A set of maps, line graphs, and boxplots, as well as a probability density function. Using the "think aloud method" and a "marked protocol" [38], we identified features that hindered as well as those that facilitated understanding of these visualizations.

Like any study, ours had limitations. First, even though our sample size was large enough for qualitative analyses about which misunderstandings might occur for diverse audience members of communications about climate projections [33], it was not large enough to indicate the prevalence of those misunderstandings, or how they varied by specific groups. Second, although we selected commonly used visualization formats, we only selected visualizations that displayed projections for expected rainfall in the UK, and did not include other common visualization formats. Third, we limited our focus to target audiences in the UK.

However, we think that our findings have some important implications for re-designing visualizations. Based on our findings and literature identified in the wider field of risk communication, we now discuss several strategies for re-designing such visualizations for audiences who need to make adaptation decisions. Some of these strategies are related to the content of visualizations, while others concerned editorial issues around visualizations. Following this research, strategies were partially implemented into visualizations used in the United Kingdom Climate Projections launched in 2018 [7]. In addition, Figures 5–7 present re-designed visualizations based on the below strategies.

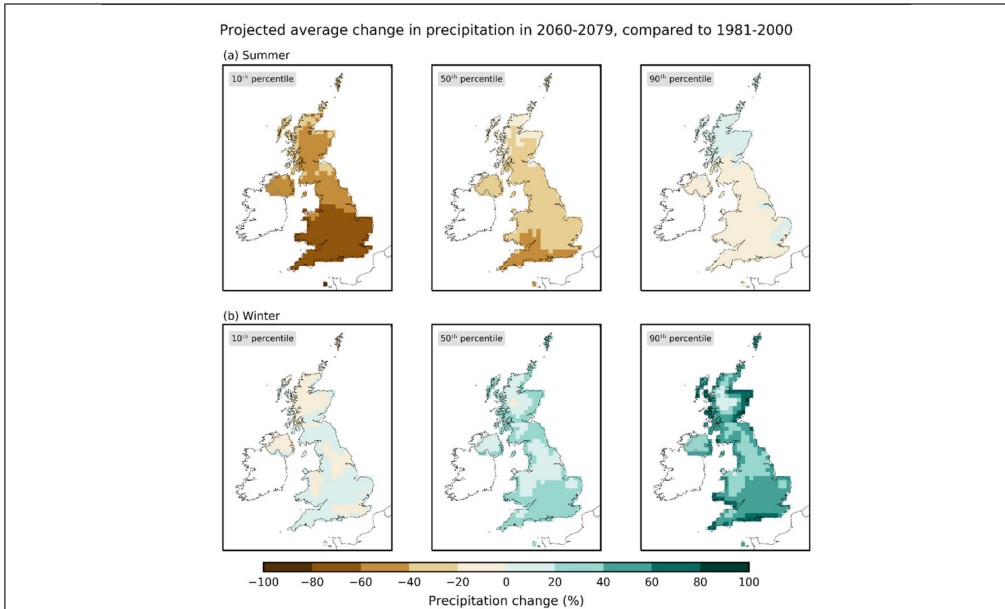

Projected average precipitation change (in %) over the UK for 2060–2079, compared to a 1981–2000 average, in (a) summer and (b) winter. The projections are probabilistic: The 50th percentile is the most likely projected change. Changes smaller than the 10th percentile and greater than the 90th percentile are very unlikely. Maps are for Representative Concentration Pathway (RCP) 8.5, representing a high greenhouse gas emissions scenario. UKCP probabilistic projections reflect a 25 km spatial resolution. UKCP probabilistic projections are described in greater detail in Fung et al. (2018).

**Figure 5.** Map and associated caption displaying summer and winter rainfall change in the United Kingdom, as re-designed by the authors, based on study findings and strategies in Table 5. Note: We re-designed the maps (Figure 1) based on participants' feedback and suggested strategies.

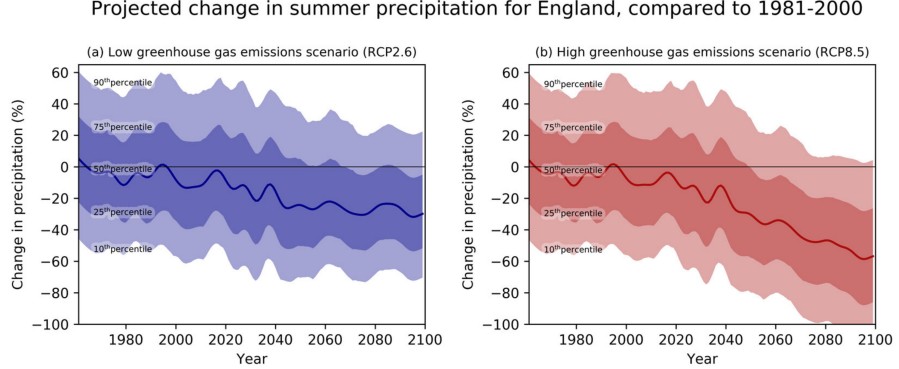

Projected changes (in %) in summer precipitation for England, compared to a 1981–2000 average. The projections are probabilistic: The 50th percentile is the most likely projected change. Changes smaller than the 10th percentile and greater than the 90th percentile are very unlikely. Changes at the 25th and 75th percentiles are also shown. **(a)** shows projections for a low greenhouse gas emission scenario involving strong mitigation action, namely Representative Concentration Pathway 2.6 (RCP 2.6, in blue). **(b)** shows projections for a high greenhouse gas emissions scenario involving little mitigation, namely Representative Concentration Pathway 8.5 (RCP 8.5, in red). UKCP probabilistic projections are described in greater detail in Fung et al. (2018).

**Figure 6.** Line graphs and boxplots, and associated caption displaying summer rainfall change in England, as re-designed by the authors, based on study findings and strategies in Table 5. Note: We re-designed the line graphs (Figure 2) based on participants' feedback and suggested strategies. To reduce complexity, the boxplots were dropped.

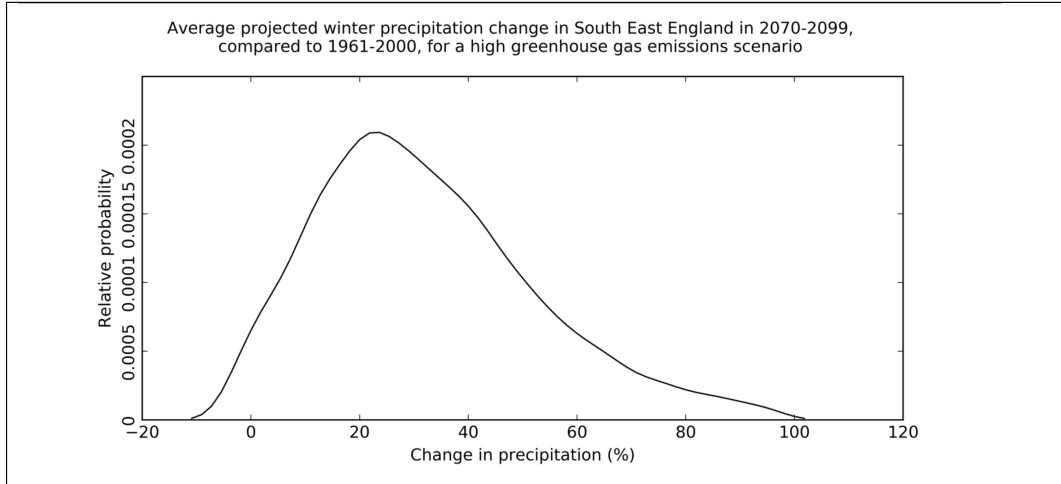

Projected changes in winter precipitation (in %) for South East England from the UKCP Probabilistic Projections. The curve shows the relative probabilities of average projected changes. Projected changes are for 2070–2099 compared to a 1961–1990 average for a high-emissions scenario. Relative probabilities from the probabilistic projections are described in Fung et al. (2018).

**Figure 7.** Probability density function displaying precipitation change in South-East England, as re-designed by the authors, based on study findings and strategies in Table 5. Note: We re-designed the Probability Density Function (Figure 3) based on participants' feedback and suggested strategies.

### 4.1. Content-Related Strategies

Participants pointed out that statistical concepts, including those describing uncertainties such as probabilistic estimates, were difficult to understand. They asked for a caption to incorporate further explanations. Other studies have shown that simple visual aids, such as likelihood distributions, may help to improve recipients' understanding of the shape of the underlying probabilistic distribution [42]. We also recommend explaining statistical features, such as probabilistic thresholds and the nature of underlying distributions ([11,42], strategy 1, Table 5).

While participants asked for further explanations about time intervals used as past and future reference periods, only two mentioned comparisons to the widely used "pre-industrial times". Target audiences may thus benefit from a justification of past and future time periods used, as well as consistency in time intervals used.

Furthermore, participants struggled to understand relative change communicated in percentages. They noted that a baseline or reference point was missing. This made it difficult for them to evaluate the magnitude of change displayed. Studies about the relative probability of rainfall [43] as well as health risk communications had similar results. Here, the magnitude of an effect due to a medical intervention was easier to understand when this was presented in natural frequencies including a baseline group, rather than as a relative change in percentages [29,44]. This format allows evaluation of the relationship of a part to the whole and helps in alleviating inflated risk perceptions [45,46]. We thus recommend communicating absolute change in the original unit (such as mm), or, where appropriate, to relate change in percent to a baseline value in the original unit (strategy 3).

Across visualizations, participants mentioned unfamiliar language and acronyms (Tables 2–4). Research from the health field indicates that audiences clearly benefit from simple, clear, and consistent titles and captions [31]. They also benefit from content described with shorter words and shorter sentences, reflecting reading skills of 5th–6th graders [41], as well as with language that is intuitive and familiar to them [31]. Our recommended strategy is thus to simplify language (strategy 4).

**Table 5.** Strategies for designing climate projection visualizations.

| Strategy |
| --- |
| **Content-Related Strategies** |

1. Carefully explain underlying statistical concepts for describing uncertainty, such as probabilistic estimates.

2. Use time periods of similar length in comparisons of a past baseline with future change, combined with an explanation.

3. Communicate absolute change where appropriate: Either specify the reference class for communicating relative change in percentages, or communicate change in the original unit rather than in percentage change.

4. Simplify language by using short words and short sentences, avoiding acronyms, and using language that is familiar to target audiences.

5. Simplify visualizations by focusing on one main message, and also explain that main message in the caption. Split complex data into smaller chunks, and make relationships between these clear.

6. Provide basic training in climate science to target audiences, which allows them to understand necessary terminology and statistical concepts.

**Editorial Strategies**

7. Use the same clear terms in the visualization and caption.

8. Reduce spatial distance between similar elements in visualization and caption by placing text into the visualization or adding graphical elements to the caption

9. Use large-enough size and resolution, especially for relevant features.

Note: Following this study, strategies 1, 2, 4, 5, and 7 were adopted for re-designing visualizations in the United Kingdom Climate Projections 2018 [7].

In addition, participants were overwhelmed by the amount of information in the displays. Literature from both health and climate suggested that focusing on one main message and reducing complexity facilitates understanding [11,32,33]. Indeed, complicated displays may be a barrier to the uptake of risk information [32]. Using a "less-is-more" approach when designing a visualization [25] may thus support information uptake. However, given the complexity of climate data and the needs of some target audiences for detailed and technical information, this may not always be appropriate. Other authors thus recommend displaying data in smaller chunks and making clear how these chunks relate to each other ([11], strategy 5).

Audiences may further benefit from basic training for using communications [27], such as those about climate projections. This may include unfamiliar terminology as well as underlying statistical concepts, such as probabilistic estimates (strategy 6).

*4.2. Editorial Strategies*

We identified several editorial strategies that we recommend for use in communicating the content and the complexity of climate visualizations.

Participants pointed out that for the presented climate projection visualizations, the accompanying captions differed in language used for describing the same elements (such as "business-as-usual" used interchangeably with "RCP 8.5"). Furthermore, terminology may have been confusing because it included vague or outdated terms, such as "business-as-usual", for describing high-end emission scenarios [37]. We thus recommend using the same terminology in the climate projection visualizations and accompanying captions, and to ensure that terminology is clear (strategy 7).

Participants struggled to match parts of the caption to elements of the visualization because they had to search for pieces of information in the caption ("spatial contiguity effect"; [11]). We recommend

reducing the spatial distance between related elements of visualizations and the accompanying caption (strategy 8). This may involve placing some text in the visualization, including in text boxes, or adding graphical elements to captions [11]. Similarly, participants mentioned that colors were distinct on the legend, but that it was challenging to map them onto the visualization. Colors for different degrees of change must be sufficiently distinct to be mapped onto the visualization.

Finally, important features of climate projection visualizations were too small. Research in cognitive sciences indicates that attention can be attracted by increasing the size of relevant visualization features [11]. We thus recommend increasing the size of relevant features and using large-enough sizes and resolutions (strategy 9).

## 5. Conclusions

Our findings and associated strategies need to be considered within the broader literature on risk and uncertainty communication in the climate domain and related disciplines. This literature suggests the presentation of "gists"—the most important information—to non-experts, rather than a high level of detail, which may overwhelm target audiences [31,32,46]. Based on our findings, we can confirm that often, when it comes to uncertain climate projections presented visually, "less is more". In other words, simple visualizations may help many audiences to process information about climate hazards and associated uncertainties. Strategies as suggested in this paper aim at identifying how to communicate and simplify the most relevant elements of such uncertain climate projections where appropriate. As in our case, they may then be adopted in communications similar to the United Kingdom Climate Projections 2018 and help to design communications that are understandable for audiences with limited backgrounds in climate science.

Our findings allow several avenues for further empirical research. First, interviews as reported in this study provide insights into which challenges may arise when communicating inherently uncertain climate information, but not how often they will arise. Surveys with larger samples would be needed to identify the prevalence of specific communication problems. Additionally, randomized controlled trials with larger samples would be needed to assess how effective our strategies are for reducing the identified communication problems. These need to control for differences between various members of these target audiences, such as in numeracy [47], levels of education, or their background in climate sciences. They also need to experimentally test strategies identified here, such as different explanations for statistical concepts (strategy 1) or different numerical formats for communicating relative change (strategy 3).

Second, visualizations re-designed based on insights from interviews and surveys need to be experimentally tested, similarly to communications from other disciplines, such as health [32].

Third, studies need to investigate what types of decisions target audiences associate with the use of communications about climate projections [4]. This may include how different wordings of decision outcomes influence perceived costs, risks, and benefits related to different decision options and, consequently, adaptation [4]. This also includes a geographic resolution that allows target audiences to assess the impact of their decisions.

Fourth, future studies need to include an assessment of how target audiences access information about climate projections. They may use tailored tools that include uncertainties relevant for their decisions, wording they are familiar with (the UK-based Environment Agency provides examples [48]), and explanations of both the underlying statistical concepts (such as probability density functions) and the specific climate models used. Stakeholders may also benefit from science reports targeting their specific sector [49]. Finally, uncertain climate communications use a much wider range of visualization formats than those studied here, such as various maps, line graphs, boxplots, infographics, or roulette wheels, just to name a few [9,10,50]. These need to be tested with target audiences [51].

Overall, communications about climate projections need to be adapted to stakeholders' needs and levels of understanding. These needs may not necessarily match what scientists deem as relevant when they design communications about uncertain climate projections for non-scientific audiences [4,33]. Such audiences may also lack a background in statistics, including knowledge about

probabilistic estimates. Empirically tested communications about climate projections substantially facilitate adaptation to a changing climate.

**Supplementary Materials:** The following are available online at http://www.mdpi.com/2071-1050/12/7/2955/s1, Table S1: Interview protocol as used in the reported study.

**Author Contributions:** Conceptualization, A.K., W.B.d.B., F.F., and J.L.; formal analysis, A.K.; funding acquisition, A.K., W.B.d.B., and J.L.; methodology, A.K., W.B.d.B., and J.L.; project administration, W.B.d.B. and J.L.; resources, F.F. and J.L.; supervision, W.B.d.B., A.T., and J.L.; visualization, F.F.; writing—original draft, A.K.; writing—review and editing, A.K., W.B.d.B., F.F., A.T., and J.L. All authors have read and agreed to the published version of the manuscript.

**Funding:** This research was funded by M2D, the Models to Decision Network (grant number M2DPP 022) and the Met Office UK, Strategic Priorities Fund (grant number CR19-6a. Wändi Bruine de Bruin was supported by the Swedish Riskbankens Jubileumsfond's program on "Science and Proven Experience" and the Center for Climate and Energy Decision Making (CEDM) through a cooperative agreement between the National Science Foundation and Carnegie Mellon University (SES-0949710 and SES-1463492).

**Acknowledgments:** We thank Suraje Dessai for comments on an earlier version of this manuscript, and Justin Hopper for editing the manuscript.

**Conflicts of Interest:** The authors declare no conflict of interest. The funders had no role in the design of the study, in the collection, analyses, or interpretation of data, in the writing of the manuscript, or in the decision to publish the results.

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
