# Peer review of "Visualizations of Projected Rainfall Change in the United Kingdom: An Interview Study about User Perceptions"

_sustainability, doi:10.3390/su12072955_

Round 1

Reviewer 1 Report

The present manuscript is to study user perceptions in responding to various climate model visualizations using some example models and survey data. Better communication with the public for the issues of climate change is needed, but its practice is still very much for science communities rather than non-major public communities. For that reason, the present manuscript has a value to look into, especially its effort for quantitative analysis is very meaningful. The only weakness is that the data size would be a bit small (24 total) to make some conclusive arguments. Since the present manuscript would be a beginning study for further investigation with more survey data and testing in the future, I would like to still accept the manuscript for publication. Also, I would like to see more enhanced survey conditions. For example, when the Authors showed Figure 1 the interviewees, how much information was given to them? Depending on what level of information was given, the level of understanding would be very different. Probably Authors can add more descriptions of such conditions. As the authors made a list of suggestions, it would be interesting to see how such changes would make a difference in terms of user perceptions.  I would like to suggest minor corrections though – 1. Figure 1 missed (a), (b), and (c). Figure 1 (c) font is too small. 2. Tables need to have horizontal lines to distinguish each item.

Reviewer 2 Report

General

The authors correctly state that it is important to understand how target audiences perceive climate visualizations. The choice of visualizations is suitable, i.e., maps, time series, and probability density functions because these are commonly used in climate research and outreach. Other types could be discussed for completeness.

The paper lacks communication and editorial correctness, such as acronyms that are not spelled out when first used, figure captions that are not properly labelled (a,b,c in figure one), or not sufficiently comprehensive with a mismatch to the figure (figure two). Considering the topic, this is surprising.

The methods require further discussion and detail, such as how the participants were chosen, why so many types of participants were interviewed (no specific audience), why the older examples from the UK climate projection project were used, and why the rainfall variables were chosen, etc. Some barriers to communication mentioned are not climate communication barriers, but editorial problems that would be relevant to any paper. The results therefore seem to be common sense and editorial correctness.

In general, even though the topic of the paper is worthwhile, the paper has several gaps, errors, and other problems which require addressing. Several of the problems hindering understanding of the figures, such as use of acronyms and labelling, seem to result from lack of editing and should be discussed as such. They are not really a climate communication problem. For example, conventionally acronyms are spelled out where first used in the text.

Specific

Figure 1 The caption must include the source of the figures. Labels a to c are missing from the figures. This is an editorial omission, not a climate communication problem. For sufficient evaluation, the captions and figures should be the same as provided to the people interviewed. From the responses, this does not seem the case. Also, have the figures and captions been changed from the UK source? Explain. Also, explain the use of the figures in the interviews. Are they meant to stand alone, or be used in conjunction with text for explanation of the acronyms and concepts?

Lines 96 etc Acronyms should be spelled out where first mentioned in this paper. This first example is UKCP18, followed by several others (e.g., RCP2.6 line 118, SRES A1B line 121). These four RCP scenarios should be briefly explained here too. For such a general audience, it is not reasonable to assume understanding of the RCPs.

Line 97 The visualizations selected are mentioned to be partially inconsistent etc. Why? Why were preliminary figures used, not the more recent ones? Explain.

Line 118, 112 etc Why continue to use the term “business-as-usual” when the criticism of it is mentioned earlier (line 98). Change to something like “High emission scenario” as it is more suitable.

Section 1.4 introduces the study. The reasons for selecting these types of visualizations should be provided as well as the reasons for selecting precipitation data.

Section 2 Materials and Methods: Limitations of the research, such as the number of interviews should be discussed and the use of the visualizations. Also, the interviews are from a wide variety of stakeholders with very different backgrounds. The audience is often considered in communications and the visuals should be targeted to the audience.

Indicate how the 24 participants were chosen. Discuss why were they chosen from such a wide range of characteristics. This is an important part of the methods that is missing as well as a discussion of the limitation of the number of participants. Indicate whether the participants were meant to represent a certain population, apart from “general” users.

Table 1 is useful, but the acronyms and the scale need to be described for best communication. The scale is described in the text, line 166, but should be in the table caption also. Again, the authors did not seem to heed their own recommendations.

Section 3 Results

The caption for Figure 2 mentions “median aggregated helpfulness” but the figure includes much more information. Therefore, the caption is incorrect and should be more comprehensive to match the figure (more than just helpfulness and the median). The note below the caption should be included in the caption. Why are outlier dots shown for the lower part of the scale, but not the upper portion, and not for the line graph at all? This is unusual and should be corrected or explained.

Line 192: this is useful knowledge that the maps and their captions are relatively helpful, and more helpful than the other two types. This should be included in the discussion.

Line 216 Change “Reaction Concentration..” to “Representative Concentration…”

Line 219 This description of probabilistic estimates should be in the caption of Figure 1 along with any other information used in the captions presented to the reviewers. That is, Figure 1 and its caption should be the same as used in the interviews for proper assessment by the readers of this paper.

Section 3.2.1 Map features that hindered understanding. Regarding acronyms and jargon, this critique seems unfair.  It is editorial convention to explain acronyms and jargon where they are first used in the text. If the visualization is meant to stand-alone, these would be explained in the caption or notes with the figure. Therefore, this problem is editorial and should not be considered in this section if dealt with properly.

Line 255 Again, the omission of the letters on the map is not a climate figure problem, but is an editorial error and common sense. It should be treated as such and not labelled as hindering understanding of a climate map. Such errors are usual fixed before publication, and should be noted as such. A recommendation is that figures to be used for assessment require careful editing before use to avoid such problems. This is true for any communication.

L308 If the meaning of “Global Climate Model” is not clear, this should be supported by the text or another figure. Again, explain if the figures are meant to be stand alone or not. The use of GCMs is a basic aspect of understanding future climate change. This means that the interviewees require much more basic training in climate change concepts.

Line 313 Again, Figure 1 should have the captions used in the interviews for adequate interpretation of the results by the reader. Observed data coming from the models does not make sense here.

Line 320 Confusion about percentiles is odd as percentages are usually taught in primary school along with fractions. Can this be explained?

Line 329 re “climate scenarios (RCP2.6…). The RCPs are emission scenarios and are used to create climate scenarios. Reword.

Line 348 Again, missing labels should have been fixed at the editorial stage and are not really a barrier if this basic labelling was not missing.

Section 3.3 re probability density function. This rating is not surprising considering that the audience would need some background in statistics for understanding.

Line 371 use the word “fewer” rather than “less”.

Line 387 The legend box on Figure 1 for the PDF looks as if it was for the researchers and not users. This type of information in the legend belongs written out suitably in the caption. Again, this seems to be editorial oversight by the authors. Then such problems as confusion over the word “high” would not be as likely.

Line 388 This barrier to understanding would not occur if the text, such as the caption, explained the acronyms and jargon, especially if the figure is meant to be used without additional text. This is a standard feature of papers. If people did not understand PDF, as indicated, then the entire figure would not be understood.

A contrast and comparison of the three types of visualizations would be useful for the results, but is missing. This discussion would be beneficial.

Section 4. Discussion

Line 404 Delete the future tense “will” as the need to adapt is in the present as well as the future.

Line 412 The term “marked protocol” was not in the methods and requires description there.

Line 432 Change “hole” to “whole”. This is another indication of editorial lapses by the authors.

Line 434 The most relevant unit for example here is “mm” for precipitation, not mm2.

Line 440 Simplified language is a suitable strategy. However, for more complex science such as climate change, definitions and basic descriptions are also warranted. Something like a basic climate change course seems to be beneficial and recommended for users as well. Simplified language may not be suitable, depending on the use of the information and the user. This may also be a strategy to help with the overwhelming amount of information as well. Explain.

Line 451 Why were outdated terms and figures used in this study? Describe why the more recent figures and terms not used.

Line 463 re features of the figures being too small. This may be an editorial over-sight as well, as standards exist for font sizes for different aspects and may not have been followed for the presentation of the figures.

Table 5 item 3 “Communicate absolute change”. This may not always be appropriate, especially in comparing precipitation amounts.

Table 5 is a good idea to have these strategies displayed. These strategies seem to be editorial and communication common sense, not new findings.

Note on Line 468 If some of the strategies were used already, explain why the re-designed figures were not used in the study. Explain why only some of the strategies were used.

No discussion of other types of visuals exists. Explain.

Section 5 Conclusions

Line 480 Re simple visualizations being better. Note that this really depends on the audience. The design and types of visuals should be targeted to the audience, as possible. Some audiences would require more technical and complex information. Include a caveat. This is hinted at in line 490. Audience needs and expertise must be considered with more targeted design of figures.

Many of the strategies summarized in Table 5, e.g. simplify figures, simplify language, using clear terms seem to be common sense.

The avenues for further research are generally appropriate.

Reviewer 3 Report

Thank you to the authors for submitting a well written piece of work.

This piece isn't novel, however, it does offer an empirical example of testing visualisations with publics. 

My only caveat is there are 2 key elements that were not addressed in the introduction, methods or limitations.

That is that climate visualisations inherently require apriori knowledge. 

In addition, they are supplemented to the reader with text. Within the study here it appears the visualisation was to stand along.

Further, scientific visualisations are very specific, in order to communicate primarily with other scientists who have said knowledge of required underpinning science. Others may not. 

Rather than changing scientific visualisations to cater to a public without scientific experience, it is possible to adapt the graphs, visualisations etc into an 'infographic'. These types of vis by nature are created for a general public, rather than specialists. 

Although there are elements and principals of good data vis (which I believe this paper should reference, yet doesn't explicitly do so in the current version), even good data vis of climate science is difficult to read. The tenants such as consistency, colours, emphasis, acronyms etc are well researched (see literature on not only climate data vis, also semiotics). 

In order to strengthen this work, I suggest the authors introduce some of these elements and note further some of the limitations of the work.

It is admirable that this work is applied and many suggestions have been taken up by government actors. I suggest this outcome is highlighted further within the conclusions. 

Round 2

Reviewer 2 Report

The authors have completed acceptable revisions of this next version and it is improved. 

I recommend that next time the authors do such work that the figures be carefully edited by an editor before the research work to avoid these more easily avoidable errors. This would also make the research even more useful.